# Influence of Saharan Dust Intrusions on Respiratory Medication Dispensing

**DOI:** 10.3390/medsci13040234

**Published:** 2025-10-20

**Authors:** Ruperto González-Pérez, Ainhoa Escuela-Escobar, Mario A. González-Carracedo, Paloma Poza-Guedes

**Affiliations:** 1Allergy Department, Hospital Universitario de Canarias (HUC), 38320 San Cristóbal de La Laguna, Spain; pozagdes@hotmail.com; 2Severe Asthma Unit, Hospital Universitario de Canarias, 38320 Tenerife, Spain; 3Instituto de Investigación Sanitaria de Canarias (IISC), 38320 Santa Cruz de Tenerife, Spain; 4Instituto Universitario de Enfermedades Tropicales y Salud Pública de Canarias (IUETSPC), Universidad de La Laguna (ULL), 38200 San Cristóbal de La Laguna, Spain; aescuela@ull.edu.es (A.E.-E.); mgonzalc@ull.edu.es (M.A.G.-C.); 5Genomics and Health Group, Department of Biochemistry, Microbiology, Cell Biology and Genetics, Universidad de La Laguna (ULL), 38200 San Cristóbal de La Laguna, Spain

**Keywords:** Saharan dust intrusion, PM_10_, asthma, COPD, short-acting beta-agonists, inhaled corticosteroid–long-acting beta-agonist

## Abstract

Background: Saharan dust intrusions (SDIs) are associated with poor air quality and adverse respiratory outcomes. However, their impact on real-world inhaler utilization remains insufficiently characterized. We aimed to examine the association between SDI and the dispensing of short-acting beta-agonists (SABA) and inhaled corticosteroid–long-acting beta-agonist (ICS–LABA) combinations in the Canary Islands, Spain. Methods: Pharmaceutical sales data for SABA and ICS–LABA were collected from 60 pharmacies in Santa Cruz de Tenerife (TF) and Las Palmas de Gran Canaria (GC) between June 2017 and May 2022. SDI days were identified based on daily PM_10_ concentrations > 40 µg/m^3^ from the regional air quality monitoring network. Linear regression models evaluated associations between drug dispensations and SDI presence, frequency, and intensity, adjusting for seasonality (winter vs. summer). Results: Over 60 months, SABA sales were 14.8% lower in TF compared with GC, while ICS–LABA sales were 10.9% higher. SDI presence was associated with significantly higher ICS–LABA dispensations in both provinces (+5.7% in TF, +10.2% in GC), whereas no association was found for SABA. ICS–LABA sales correlated weakly but significantly with both SDI frequency and PM_10_ levels. Seasonal analysis revealed stronger effects in winter, with ICS–LABA dispensations increasing by 14.3% (TF) and 9.6% (GC) during SDI months. For SABA, seasonal differences were independent of SDI exposure. Conclusions: SDIs in the Canary Islands are independently associated with increased dispensing of ICS–LABA maintenance therapy, particularly during winter months. Dispensing data offer a valuable population-level indicator of respiratory impact from natural airborne pollution and support the integration of environmental alerts into preventive respiratory care strategies.

## 1. Introduction

The exposome, first introduced by Wild [1], refers to the totality of environmental exposures encountered across the lifespan and their biological interactions. This concept complements genomic research by offering a broader framework to understand complex, multifactorial diseases [2,3,4]. Among the diverse external exposures, those related to air quality are particularly critical. In this context, desert dust intrusions—such as those originating from the Sahara—represent episodic yet substantial components of the atmospheric exposome. These events contribute to sharp increases in particulate matter and transport not only mineral particles but also microorganisms, allergens, and toxic compounds, with significant implications for human health [5,6].

Desert dust intrusions are large-scale atmospheric phenomena driven by strong winds and arid conditions, capable of lofting fine and coarse particles into the atmosphere and carrying them thousands of kilometers. As a result, air quality can deteriorate far from the emission source, affecting populations across entire regions [7,8]. While dust has traditionally been considered a natural element of Earth’s geochemical cycles, the rising frequency and intensity of dust storms—exacerbated by land degradation, desertification, and climate change—have transformed them into an emerging public health challenge [9].

One of the most relevant consequences of dust exposure is its impact on respiratory health. Fine particulate matter (PM_10_ and PM_2.5_), abundant during dust events, can penetrate the lower respiratory tract and trigger inflammation, oxidative stress, and bronchial hyperresponsiveness [10,11]. Epidemiological studies have consistently linked dust episodes with higher rates of asthma and chronic obstructive pulmonary disease (COPD) exacerbations, emergency department visits, hospital admissions, and mortality, particularly among vulnerable populations [12,13,14]. Other natural hazards such as wildfires and volcanic eruptions also produce airborne pollutants—fine particulates, carbon monoxide, irritant gases—that contribute to respiratory morbidity. In recent years, wildfires have become more frequent and intense, causing episodes of severe air pollution across wide geographic areas [15,16]. In addition, desert dust can carry biological material including allergens, bacteria, and fungi, which may further aggravate symptoms in patients with asthma or immunologic susceptibility [17,18].

Real-world data increasingly support the notion that these natural airborne pollution events are followed by short-term surges in inhaled medication use, particularly short-acting beta-agonists (SABAs), inhaled corticosteroids (ICS), and combination maintenance therapies such as ICS–LABA and long-acting antimuscarinics [19,20]. In regions affected by Saharan dust—such as Southern Europe and the Middle East—dispensing records have shown a rise in rescue and maintenance inhaler use within days of exposure peaks [21,22]. Similar trends have been observed during wildfire seasons in North America and Australia, where pharmacy sales of SABAs increase in parallel with acute respiratory distress [23,24]. These patterns highlight both the clinical burden of environmental episodes and the utility of inhaler dispensing as a proxy for population-level respiratory impact. Moreover, inhaler utilization trends reveal disparities in access and preparedness: during periods of elevated pollutant exposure, patients with asthma or COPD may rely more heavily on rescue inhalers, especially when access to routine care is disrupted. This raises important questions regarding the adequacy of preventive strategies, the timeliness of public health alerts, and the resilience of healthcare systems during environmental emergencies.

Previous research suggests that the respiratory impact of air pollution may be more accurately captured through patterns of medication use than by symptom reporting or exacerbation counts [25,26]. Using Spain’s universal prescription registry, we investigated Saharan dust intrusions (SDI) in the Canary Islands, a setting uniquely suited to this question given its frequent exposure to SDI [27,28] and elevated prevalence of asthma and COPD [29,30,31]. Located in the eastern subtropical Atlantic, the Canary Islands form a Spanish volcanic archipelago situated approximately 100 km off the northwest coast of Africa (27°–29°N, 13°–18°W) [32,33]. This geographical proximity exerts a major influence on the local climate, which is characterized by mild, low-variability temperatures but is frequently affected by the *Calima*—an episodic advection of mineral dust-laden air from the Sahara Desert. On average, these dust intrusions affect the Canary Islands on approximately 24 days per year, with a mean duration of about 1.8 days per episode [34]. The most intense and prolonged events typically occur during winter and early spring (October to March), when low-altitude transport markedly increases particle concentrations at ground level [35]. This critical period of peak dust exposure often coincides with the seasonal circulation of viral respiratory infections (such as respiratory syncytial virus) and may precede the onset of the main pollen season, creating a complex, high-risk interval that has been associated with increased emergency hospital admissions for asthma and other respiratory diseases across the archipelago [36,37]. The objective of the present investigation was to determine whether SDI are associated with changes in dispensing of SABA reliever therapy and ICS–LABA maintenance therapy.

## 2. Materials and Methods

### 2.1. Definition of ICS-LABA and SABA Sales

Sales data (number of units dispensed) of Inhaled Corticosteroids plus Long-Acting Beta-Agonists (ICS-LABA) and Short-Acting Beta-Agonists (SABA) were retrieved from the Sanibrick digital health platform (accessed on 22 August 2022), which aggregates anonymized monthly records of pharmaceutical sales from community pharmacies across Spain for epidemiological analysis [38]. Each Sanibrick represents a data aggregation unit that includes several pharmacies within the same locality or nearby areas. Sixty out of 79 (75.94%) Sanibricks with continuous data reporting throughout the study period were included, without further selection. Each Sanibrick corresponds to multiple Community Pharmacies, resulting in a total of 763 pharmacies officially distributed across the Canary Islands (390 in Santa Cruz de Tenerife (TF), and 373 in Las Palmas de Gran Canaria (GC)) [39].

The data timeline was established from June 2017 to May 2022 (60 months of follow-up). The monthly sum of sales for each drug was calculated, resulting in 60 monthly observations for each province. Sales were assigned to the month following dispensing to better reflect the effective consumption period.

### 2.2. Definition of SDI

Days with SDI were identified considering the daily concentration of Particulate Matter with a diameter ≤ 10 µm (PM_10_), retrieved from the Canary Islands Air Quality Control and Monitoring Network [40]. Daily mean PM_10_ concentrations were calculated from 30 and 21 monitoring stations in TF and GC, respectively, and the timeline included the same period of 60 months defined above. The presence of SDI was declared when the mean PM_10_ concentration exceeded 40 µg/m^3^, a threshold at which air quality is classified as “regular” according to the criteria defined by the Canary Islands Air Quality Control and Monitoring Network [40].

Since different SDI atmospheric scenarios take place in the Canary Islands depending on the season, the effect of seasonality was also considered. During the winter (1 October to 31 March), Saharan dust is mainly carried by low-level continental African trade winds (*Harmattan* winds), reaching altitudes of less than 700 m above sea level, and mostly affecting urban populated areas. However, in the summer (1 April to 30 September), dust is mainly transported by the northern branch of the high-altitude Saharan Air Layer, reaching higher altitudes where urban-populated regions are almost absent [41,42,43].

### 2.3. Statistical Analysis

Data analyses were performed using R software (version 4.3.2; R Core Team, 2023) within RStudio (version 2023.09.1+494; Posit Software, PBC). Outliers were identified using the interquartile range (IQR) criteria (datapoints 1.5 × IQR above the third quartile or below the first quartile were excluded). Outliers were identified independently for each analysis group and removed prior to statistical testing. Data normality was assessed using the Kolmogorov–Smirnov test (*n* > 50). Linear regression models were applied to evaluate the association between ICS-LABA or SABA sales with the presence/absence of SDI, the total number of days with SDI per month, and the monthly average PM_10_ concentration. Additionally, linear regression models were adjusted for seasonality (winter or summer), defined as described above. Statistical significance was declared for those comparisons with *p*-value < 0.05, and correlations were evaluated with the Pearson’s coefficient (*r*).

### 2.4. Ethical Approval and Consent to Participate

The study was conducted according to the guidelines of the Declaration of Helsinki and approved by the Institutional Ethics Committee of CEIm Hospital Universitario de Canarias, Tenerife, Spain with the reference number P.I.-2017/72 on 30 October 2017. The study was conducted using aggregated pharmaceutical sales records and publicly available air quality data therefore, no identifiable personal information was collected, and all data were anonymized at the pharmacy level. The requirement for informed consent was waived, as the study involved only secondary, non-identifiable data and posed no risk to individuals.

## 3. Results

### 3.1. Effect of SDI over SABA and ICS-LABA Sales in the Canary Islands

The monthly sales of SABA and ICS-LABA during the 60-months follow-up was studied in the two provinces of the Canary Islands (Table 1).

Results showed that SABA sales were 14.8% lower in TF, compared with GC (Figure 1A), while ICS-LABA sales were 10.9% higher (Figure 1B).

These results suggest a greater adherence on rescue medication in GC, while TF showed slightly higher use of maintenance therapy, and possibly better adherence to preventive treatment, pointing to differences in disease management practices, or patient adherence. Therefore, the effect of SDI over SABA and ICS-LABA sales was independently studied in both provinces, comparing the sales that took place in months where PM_10_ reached 40 µg/m^3^ in at least one day, with the sales in months with absence of SDI. SABA sales were similar between the two groups, both in TF and GC, meaning that SDI did not affect the dispensation of rescue medication in the two provinces studied (Figure 2A). Interestingly, months with presence of SDI showed significant higher ICS-LABA sales in the two provinces (Figure 2B). In the case of GC, the increment reaches 10.2%, while in the province of TF, ICS-LABA sales were increased by 5.7%. These results indicate that SDI exposure is not translated in a higher adherence of rescue medication but is associated with slightly but significant use of maintenance therapy in both Provinces.

### 3.2. Effect of SDI Frequency and Intensity over SABA and ICS-LABA Sales

First, the effect of SDI frequency was evaluated by testing the associations between the number of days with SDI per month and the drugs sales, using linear regression models. No associations were found in the case of SABA, while ICS-LABA sales showed a significant association with the number of days with SDI per month, both in GC and TF (Figure 3A). To investigate the effect of SDI intensity, associations between the monthly average of PM_10_ concentration and the drug sales were also evaluated. Again, SABA sales showed no associations with PM_10_ average concentration, while significant associations were confirmed for ICS-LABA in both provinces (Figure 3B). Nevertheless, although associations were significantly supported, linear correlations were certainly low for both provinces, when evaluated by the Pearson’s *r* coefficient (Figure 3).

Overall, these results support a significant but weak correlation between the frequency and intensity of SDI and ICS-LABA sales, but not for SABA. This tendency was observed in both provinces independently, suggesting that the more the population is exposed to SDI, the greater the demand for maintenance therapy, but not of rescue medication. It should also be noted that pharmacy sales data did not allow differentiation between specific ICS–LABA formulations. As a result, we were unable to distinguish between ICS–formoterol inhalers, which may be used both for maintenance and reliever therapy, and other ICS–LABA combinations (e.g., ICS–salmeterol or ICS–vilanterol), which are prescribed exclusively for maintenance treatment [44,45,46]. This limitation should be considered when interpreting the ICS–LABA sales data.

### 3.3. Effect of SDI Seasonality over SABA and ICS-LABA Sales

Seasonality strongly modulates SDI in the Canary Islands. In winter, SDI mostly affects populated areas, while in summer it reaches more elevated areas, where urban centres are absent. Therefore, drug sales were compared only for the months with SDI presence, between the winter and summer seasons (Figure 4). Results showed that ICS-LABA sales were significantly higher during winter months with SDI presence than in summer months with SDI (Figure 4A). This effect was observed in both provinces independently, reaching 9.6% and 14.3% in GC and TF, respectively (Figure 4A). Moreover, when the comparison between seasons were repeated, but including those months without SDI only, no differences were observed (*p*_GC_ = 0.078; *p*_TF_ = 0.257). These results indicate that the increase in ICS-LABA sales detected in the Canary Islands as result of SDI is mostly associated with the events that take place during the winter.

When restricted to months with SDI, SABA sales remained significantly higher in winter compared with summer in both provinces (18.8% in GC and 18.9% in TF; Figure 4B). Comparable seasonal differences were also observed during months without SDI (*p*_GC_ = 0.035; *p*_TF_ = 0.039). These findings indicate that the increase in SABA sales is attributable to seasonality and not to the presence of SDI.

Finally, based on these results, linear regression models showed in Figure 2 were adjusted to include seasonality as a covariate. Findings showed that the increase in ICS-LABA sales during SDI remained significant even after adjustment by season in both provinces (*p*_GC_ = 0.022, *p*_TF_ = 0.027). This analysis showed an increase of 1014.0 ± 431.6 and 1181.7 ± 520.1 units of ICS-LABA in those months with SDI presence for GC and TF, respectively.

## 4. Discussion

Although desert dust is an inherent component of the Earth’s climate system, its intensification in recent decades—fueled by desertification, land degradation, and climate change—has emerged as a major public health concern [33]. The Sahara, which alone accounts for more than half of global dust emissions, is the predominant source, with recurrent plumes that often reach Southern Europe and particularly affect the Canary Islands [47,48].

Desert dust intrusions expose the respiratory epithelium to a heterogeneous mixture of mineral particles, metals, organic compounds, and microbial agents that compromise barrier integrity and trigger inflammatory cascades. Experimental models show that dust particles induce oxidative stress and activate the NLRP3 inflammasome, promoting release of IL-1β and epithelial alarmins such as IL-33 and TSLP [49,50]. These responses amplify type-2 and neutrophilic inflammation, enhance bronchial hyperresponsiveness, and impair mucociliary clearance, thereby worsening airway obstruction in both asthma and COPD [51,52]. Repeated or sustained epithelial injury can lower the threshold for symptoms such as wheezing, cough, and dyspnea, prompting greater reliance on inhaler therapies. Epidemiological studies linking dust exposure to exacerbations and healthcare use are consistent with these mechanistic insights and support the interpretation of inhaler utilization as a clinical marker of dust-related exposure burden within the exposome framework [53,54].

In this study, SDI in the Canary Islands were associated with a measurable increase in ICS–LABA dispensations but not in SABA use. This association persisted after adjustment for seasonal variation, indicating that dust episodes exert an independent effect on maintenance therapy utilization. By contrast, the winter increase in SABA sales was attributable to seasonal trends rather than SDI exposure. These results align with previous evidence linking natural air pollution events to heightened medication demand yet the differentiation between reliever and maintenance therapy is notable [55,56,57], While SABA use primarily reflects acute symptom relief, the rise in ICS–LABA sales suggests escalation of preventive therapy, whether initiated by patients or reinforced by clinicians. The modest but significant correlation between ICS–LABA sales and both frequency and intensity of SDI further supports this interpretation. The relatively short duration of SDI (typically 1–2 days) compared with the monthly aggregation of pharmacy sales represents a methodological limitation that may attenuate the detection of short-term associations. Future research incorporating daily or weekly dispensing data could allow a more precise temporal alignment between dust events and changes in medication use.

The absence of a parallel rise in SABA use invites two complementary explanations. First, SDI events in the Canary Islands have consistently been associated with increased morbidity from asthma and COPD, including higher admissions and exacerbations, often with a temporal lag suggestive of subacute inflammation rather than immediate bronchospasm [58]. This delayed effect may encourage greater reliance on maintenance therapy to stabilize symptom burden. Second, the growing adoption of Maintenance and Reliever Therapy (MART/SMART), in which ICS–formoterol serves as both controller and reliever, could shift symptom-driven medication demand into ICS–LABA dispensations rather than SABA, in line with current guideline recommendations [59,60]. Together, these mechanisms suggest that SDI exposure increases both the need for intensified anti-inflammatory therapy and the demand for symptom relief covered under ICS–LABA prescriptions, while leaving SABA utilization largely unaffected. Additionally, the finding of higher SABA sales in Gran Canaria should be interpreted with caution. While this could partly reflect greater adherence to reliever therapy, it may also indicate suboptimal adherence to maintenance treatment with ICS–LABA combinations, leading to more frequent exacerbations and, consequently, greater reliance on SABA medication. Therefore, SABA dispensing data alone cannot distinguish between appropriate symptom-driven use and overuse related to poor controller adherence [61,62,63]. Another drawback relates to the inability to differentiate between specific ICS–LABA combinations in the aggregated pharmacy data. In particular, ICS–formoterol inhalers can be used both for maintenance and as-needed reliever therapy (MART/SMART regimen), whereas other ICS–LABA combinations are restricted to maintenance treatment.

From a clinical and public health perspective, these findings carry several implications. Inhaler dispensing data represent a valuable real-world indicator of the population impact of environmental exposures, complementing hospital admissions and patient-reported outcomes. The stronger effect observed in winter highlights the role of seasonal atmospheric dynamics in shaping exposure risk and enhances the value of integrating environmental alerts into preventive respiratory care strategies. Moreover, the reliance on maintenance rather than rescue therapy raises important questions regarding adherence, prescribing practices, and whether SDI events prompt sustained adjustments in asthma and COPD management.

Limitations of the present study must be acknowledged. Dispensing data indicate medication supply rather than actual use and may be affected by prescribing practices, stockpiling, or pharmacy availability [64,65,66]. The absence of individual-level clinical outcomes prevents assessing whether increased ICS–LABA use improves control or reduces exacerbations. Dust composition—including allergens, bacteria, and fungi—was also not characterized, despite its potential impact on respiratory outcomes in atopic populations [67,68]. Moreover, other factors may also influence PM_10_ concentrations and inhaler dispensations, including local industrial and traffic emissions, variations in respiratory viral infections, and airborne allergen exposure. While in the Canary Islands the contribution of anthropogenic PM_10_ sources is modest compared with Saharan dust intrusions [35,69], the potential impact of these environmental and biological cofactors should be considered when interpreting the observed associations.

Future research should apply advanced time-series models (e.g., distributed lag or autoregressive) to capture delayed and cumulative effects of SDI, and link dispensing patterns with clinical outcomes such as exacerbations, emergency visits, and lung function. Incorporating chemical and biological dust analyses may further clarify exposure–response mechanisms and support targeted preventive strategies in regions recurrently affected by desert dust.

## 5. Conclusions

This study provides novel evidence that Saharan dust intrusions significantly increase the use of ICS–LABA maintenance therapy in the Canary Islands, independent of seasonal variation. In contrast, SABA utilization patterns reflected only seasonal trends, not dust exposure. These findings highlight important differences in patient and prescriber responses to environmental events and underscore the utility of pharmaceutical dispensing records as a sensitive, real-world proxy for respiratory burden. Public health systems in regions affected by desert dust should anticipate temporary increases in maintenance therapy demand, particularly during winter episodes when populated areas are most exposed. Integrating dust forecasts into early-warning systems and patient education programs may improve disease control, reduce exacerbations, and strengthen healthcare preparedness for future environmental challenges.

## Figures and Tables

**Figure 1 medsci-13-00234-f001:**
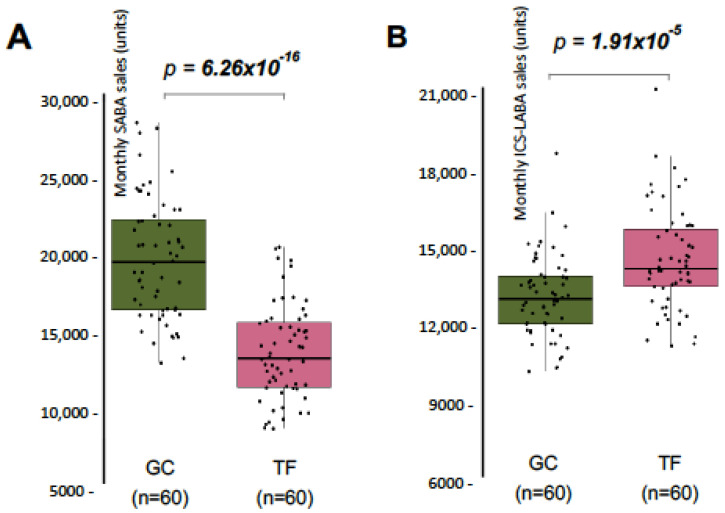
Monthly sales of SABA and ICS-LABA distributed by province in the Canary Islands. (**A**) Monthly sales of SABA for the provinces of Las Palmas de Gran Canaria (GC; green) and Santa Cruz de Tenerife (TF; pink). (**B**) Same as in (**A**), but for monthly sales of ICS-LABA. Significant *p*-values (<0.05) are highlighted.

**Figure 2 medsci-13-00234-f002:**
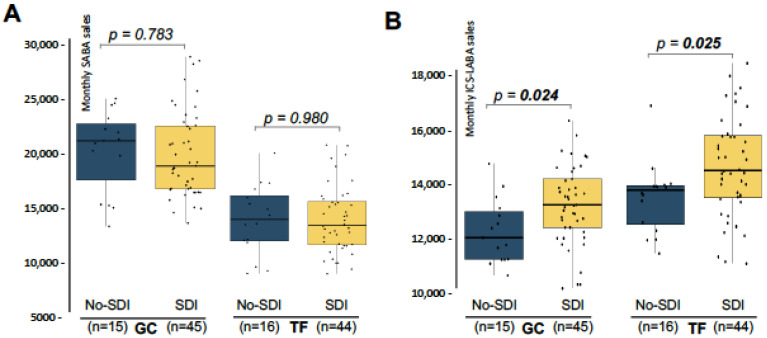
Monthly sales of SABA and ICS-LABA distribution by provinces and SDI presence. (**A**) Monthly sales of SABA (units dispensed) for the provinces of Las Palmas de Gran Canaria (GC) and Santa Cruz de Tenerife (TF). For each province, months were classified considering the absence (no-SDI; blue) or the presence (SDI; yellow) of at least one day with a PM_10_ concentration above 40 µg/m^3^. (**B**) Same as in (**A**), but for monthly sales of ICS-LABA. Statistically significant *p*-values (<0.05) are highlighted in bold. Abbreviations: n = sample size (months), SDI = Saharan Dust Intrusion.

**Figure 3 medsci-13-00234-f003:**
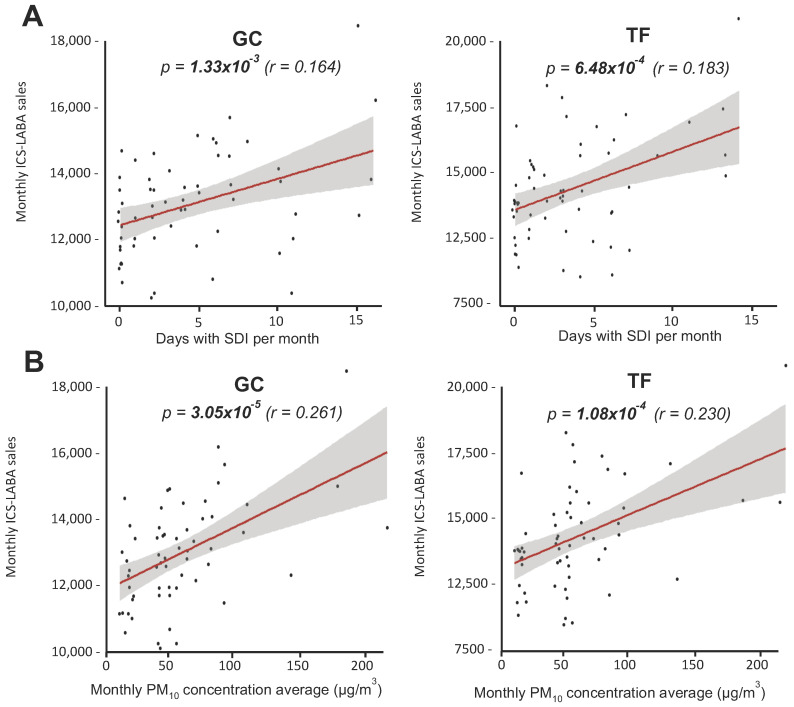
Associations between frequency and intensity of SDI over drug sales. (**A**) Results from linear regression models to evaluate associations between the number of days with SDI per month and monthly sales of ICS-LABA, both in GC (left) and TF (right). (**B**) Results from linear regression models to evaluate associations between the mean PM_10_ concentration per month and monthly sales of ICS-LABA, both in GC (left) and TF (right). Significant *p*-values (<0.05) are highlighted in bold. Abbreviations: SDI = Saharan Dust Intrusion, *r* = Pearson’s correlation coefficient.

**Figure 4 medsci-13-00234-f004:**
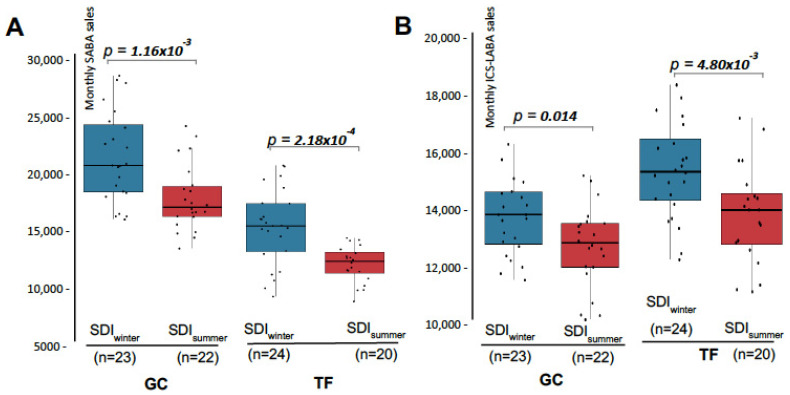
Distribution of monthly sales of SABA and ICS-LABA by provinces during winter and summer months with SDI presence. (**A**) Monthly sales of SABA (units dispensed) for the provinces of Las Palmas de Gran Canaria (GC) and Santa Cruz de Tenerife (TF). For each province, sales in winter (blue) and summer (red) months with SDI presence were compared. (**B**) Same as in (**A**), but for monthly sales of ICS-LABA. Statistically significant *p*-values (<0.05) are highlighted in bold. Abbreviations: n = sample size (months), SDI = Saharan Dust Intrusion.

**Table 1 medsci-13-00234-t001:** ICS-LABA and SABA sales in the Canary Islands, Spain. Note that the value “30 (50)” indicates that 30 out of the 60 months analyzed correspond to the winter season, meaning that 50% of the months included in the study are winter months, while the remaining 50% correspond to summer months.

Variable ^a^	n	Las Palmas de Gran Canaria (GC)	Santa Cruz de Tenerife (TF)
SABA (sales)	60	19,197.5 (16,105.8–21,945.5)	12,992.5 (11,176.3–15,276.5)
ICS-LABA (sales)	60	12,781.0 (11,817.5–13,669.5)	13,955.5 (13,253.8–15,484.3)
Season (Winter), n (%)	60	30 (50.0)	30 (50.0)
Presence of SDI, n (%)	60	45 (75.0)	44 (73.3)

^a^ Continuous variables (SABA and ICS-LABA) were summarized with the median and interquartile range (in brackets). Categorical variables (season and presence of SDI) were summarized as counts for each group and percentages (in brackets). Season includes winter months (from 1 October to 31 March). Presence of SDI was defined for those months in which PM_10_ concentrations reached at least 40 µg/m^3^ on any day. All analyses were conducted from data retrieved between June 2017 and May 2022. Abbreviations: n = sample size, SDI = Saharan Dust Intrusions.

## Data Availability

The data that support the findings of this study are available from Servicio Canario de Salud but restrictions apply to the availability of these data, which were used under license for the current study, and so are not publicly available. Data are however available from the authors upon reasonable request and with permission of Servicio Canario de Salud.

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
