# Peer review of "Influence of Saharan Dust Intrusions on Respiratory Medication Dispensing"

_medsci, 2025, doi:10.3390/medsci13040234_

Round 1

Reviewer 1 Report

Comments and Suggestions for Authors

This paper contributes to an existing body of literature on the relationship between Sahara dust intrusions and various adverse respiratory outcomes. As such, the paper is helpful, though not hugely novel. I think admissions and ED encounters are a more important outcome than inhaler dispensations. 

The paper is well written and succinct, and the design was carefully thought out and well-described. 
The strength of the study's findings is limited by the lack of association of SABAs with SDI events. I can't think of an outdoor air pollution event study that didn't find worsened air quality wasn't associated with SABA dispensations or use. Authors have attempted to explain this; sales of SABA and combo inhalers seems roughly similar in both island areas, so their second hypothesis (that many clinicians are following the GINA/SMART plan for inhaler use) seems to be the more plausible explanation.

Under Limitations, authors should indicate that other factors can affect PM10 (such as industrial and traffic emissions) and other factors beyond season (presumably through leading to more acute respiratory viral infections) and SDI can affect inhaler dispensations, such as local airborne allergens. I don't know how important such factors are on the Canary Islands. 

How were these 60 pharmacies chosen? Approximately what proportion of pharmacies on these islands was included in the study? 

MINOR COMMENTS:

Table 1: I don't understand row 3 season (winter): 30 (50). What does this mean? 
Line 189: presumably this should read: associated with slightly but significant use of maintenance therapy in both provinces.

Author Response

Answers to Reviewers

The Authors sincerely thank the Reviewers for their careful reading of our manuscript and constructive suggestions, which have helped us improve the clarity and precision of the text. Below, we address each of Reviewers´ comments point by point:

Reviewer 1:

This paper contributes to an existing body of literature on the relationship between Sahara dust intrusions and various adverse respiratory outcomes. As such, the paper is helpful, though not hugely novel. I think admissions and ED encounters are a more important outcome than inhaler dispensations. 

The paper is well written and succinct, and the design was carefully thought out and well-described. 
The strength of the study's findings is limited by the lack of association of SABAs with SDI events. I can't think of an outdoor air pollution event study that didn't find worsened air quality wasn't associated with SABA dispensations or use. Authors have attempted to explain this; sales of SABA and combo inhalers seems roughly similar in both island areas, so their second hypothesis (that many clinicians are following the GINA/SMART plan for inhaler use) seems to be the more plausible explanation.

Under Limitations, authors should indicate that other factors can affect PM10 (such as industrial and traffic emissions) and other factors beyond season (presumably through leading to more acute respiratory viral infections) and SDI can affect inhaler dispensations, such as local airborne allergens. I don't know how important such factors are on the Canary Islands. 

Answer: We appreciate the Reviewer’s valuable comment. We agree that additional factors—such as industrial and traffic emissions, respiratory viral infections, and local airborne allergens—can influence both PM₁₀ concentrations and inhaler dispensations. Although in the Canary Islands the predominant source of PM₁₀ is natural Saharan dust, with comparatively minor anthropogenic contributions [37,38], we recognize that these cofactors may contribute to background variability. We have now included a sentence in the Discussion section (limitations) acknowledging their potential influence (please, see below).

Line 404: “Moreover, other factors may also influence PM₁₀ concentrations and inhaler dispensations, including local industrial and traffic emissions, variations in respiratory viral infections, and airborne allergen exposure. While in the Canary Islands the contribution of anthropogenic PM₁₀ sources is modest compared with Saharan dust intrusions [35,70], the potential impact of these environmental and biological cofactors should be considered when interpreting the observed associations.”

How were these 60 pharmacies chosen? Approximately what proportion of pharmacies on these islands was included in the study? 

Answer: We thank the reviewer for this question and the opportunity to clarify the sampling framework. The study used dispensing data obtained from the Sanibrick digital health platform, which aggregates anonymized monthly records of pharmaceutical sales from Community Pharmacies across Spain for epidemiological analysis. Each Sanibrick represents a data aggregation unit that includes several pharmacies within the same locality or nearby areas. Sixty out of 79 (75.94%) Sanibricks with continuous data reporting throughout the study period were included, without further selection. Each Sanibrick corresponds to multiple Community Pharmacies, resulting in a total of 763 pharmacies officially distributed across the Canary Islands (390 in Santa Cruz de Tenerife, and 373 in Las Palmas de Gran Canaria). The manuscript has been modified as follows (Line 114):

From: Sales data (number of units dispensed) of Inhaled Corticosteroids plus Long-Acting Beta-Agonists (ICS-LABA) and Short-Acting Beta-Agonists (SABA) were retrieved from 29 and 31 pharmacies located in the two provinces of the Canary Islands (Spain), Santa Cruz de Tenerife (TF) and Las Palmas de Gran Canaria (GC), respectively.

To: Sales data (number of units dispensed) of Inhaled Corticosteroids plus Long-Acting Beta-Agonists (ICS-LABA) and Short-Acting Beta-Agonists (SABA) were retrieved from the Sanibrick digital health platform, which aggregates anonymized monthly records of pharmaceutical sales from community pharmacies across Spain for epidemiological analysis [38]. Each Sanibrick represents a data aggregation unit that includes several pharmacies within the same locality or nearby areas. Sixty out of 79 (75.94%) Sanibricks with continuous data reporting throughout the study period were included, without further selection. Each Sanibrick corresponds to multiple Community Pharmacies, resulting in a total of 763 pharmacies officially distributed across the Canary Islands (390 in Santa Cruz de Tenerife (TF), and 373 in Las Palmas de Gran Canaria (GC)) [39].

MINOR COMMENTS:

Table 1: I don't understand row 3 season (winter): 30 (50). What does this mean? 

Answer: Thank you for your comment. The value “30 (50)” indicates that 30 out of the 60 months analyzed correspond to the winter season, meaning that 50% of the months included in the study are winter months, while the remaining 50% correspond to summer months.

The following text has been added to the legend of Table 1 to clarify this point: “Note that the value “30 (50)” indicates that 30 out of the 60 months analyzed correspond to the winter season, meaning that 50% of the months included in the study are winter months, while the remaining 50% correspond to summer months.”

Line 189: presumably this should read: associated with slightly but significant use of maintenance therapy in both provinces.

Answer: We appreciate the Reviewer’s precise remark. The sentence has been modified as suggested (Line 207):

From: “…associated with and slightly higher use of maintenance therapy in both provinces.”

To: “…associated with slightly but significant use of maintenance therapy in both Provinces.”

Reviewer 2 Report

Comments and Suggestions for Authors

This is an interesting paper studying the impact of dust intrusions from the Sahara on asthma in the Canary Islands. 

It would of use to readers who are not very familiar with the Canary Islands to provide a brief summary of the local geography and weather patterns. For example, how often do these dust intrusions typically occur, for how long do they last, and at that time of year, particularly in relation to other periods of asthma exacerbations such as viral respiratory tract infections and exposure to pollens.

METHODS Why were ICS inhalers not included? Can there be other sources of PM10 particles other than dust from Sahara?

How long do dust incursions last? This data should be shown in the Results section. It seems that they only last for a few days, so comparing inhaler sales over a month, rather than the days of the SDI, will make it more difficult to reveal any associations between SDI and inhaler sales.

Could the associations in winter months be related to greater respiratory tract infections in Winter rather than SDI?

ICS - formoterol inhalers can be used as relievers as well as maintenance treatment for asthma, whereas other types of ICS-LABA inhalers are only used for maintenance treatment of asthma. Do you have data separating these different types of ICS-LABA inhalers?

RESULTS I am concerned about the comment that higher SABA sales in GC reflect better  adherence with reliever medication. Could it not also mean that adherence with ICS-LABA is poorer, so exacerbations requiring SABA are more likely?

Author Response

Answers to Reviewers

The Authors sincerely thank the Reviewers for their careful reading of our manuscript and constructive suggestions, which have helped us improve the clarity and precision of the text. Below, we address each of Reviewers´ comments point by point:

Reviewer 2:

This is an interesting paper studying the impact of dust intrusions from the Sahara on asthma in the Canary Islands. 

It would of use to readers who are not very familiar with the Canary Islands to provide a brief summary of the local geography and weather patterns. For example, how often do these dust intrusions typically occur, for how long do they last, and at that time of year, particularly in relation to other periods of asthma exacerbations such as viral respiratory tract infections and exposure to pollens.

Answer: We sincerely thank the reviewer for the pertinent remarks. The following text has been added in the introduction section of the revised version of the manuscript:

Line 96: Located in the eastern subtropical Atlantic, the Canary Islands form a Spanish volcanic archipelago situated approximately 100 km off the northwest coast of Africa (27°–29°N, 13°–18°W) [32,33]. This geographical proximity exerts a major influence on the local climate, which is characterized by mild, low-variability temperatures but is frequently affected by the Calima—an episodic advection of mineral dust-laden air from the Sahara Desert. On average, these dust intrusions affect the Canary Islands on approximately 24 days per year, with a mean duration of about 1.8 days per episode [34]. The most intense and prolonged events typically occur during winter and early spring (October to March), when low-altitude transport markedly increases particle concentrations at ground level [35]. This critical period of peak dust exposure often coincides with the seasonal circulation of viral respiratory infections (such as respiratory syncytial virus) and may precede the onset of the main pollen season, creating a complex, high-risk interval that has been associated with increased emergency hospital admissions for asthma and other respiratory diseases across the archipelago [36,37].

METHODS Why were ICS inhalers not included?

We appreciate the Reviewer's focus on the role of Inhaled Corticosteroids (ICS) in acute management. We must clarify that ICS monotherapy is not considered rescue medication because its primary therapeutic effect is anti-inflammatory, lacking the rapid bronchodilation necessary for immediate symptom relief during an acute exacerbation [1,2]. The recent paradigm shift toward Anti-Inflammatory Reliever (AIR) therapy utilizes ICS in a fixed combination with a rapid-onset Long-Acting Beta-Agonist (formoterol). In this context, the rapid-onset β2-agonist provides the necessary rescue function, while the ICS component simultaneously delivers controller therapy to reduce the risk of severe future exacerbations, a strategy clinically proven superior to Short-Acting Beta-Agonist (SABA) monotherapy [3.4]. Therefore, while ICS is a critical component of modern reliever strategies, it is the β2-agonist that fulfills the immediate 'rescue' requirement.

References:

  1. Barnes PJ. Inhaled Corticosteroids. Pharmaceuticals (Basel). 2010 Mar 8;3(3):514-540. doi: 10.3390/ph3030514. PMID: 27713266; PMCID: PMC4033967.
  2. Global Initiative for Asthma (GINA).Global Strategy for Asthma Management and Prevention.
  3. O’Byrne PM, FitzGerald JM, Bateman ED, et al.Inhaled combined budesonide–formoterol as reliever therapy in mild asthma.N Engl J Med. 2018;378(20):1869–77.
  4. Reddel HK, et al. Global Initiative for Asthma Strategy 2021: Executive Summary and Rationale for Key Changes. J Allergy Clin Immunol Pract. 2022;10(1):18–33.e5

Can there be other sources of PM10 particles other than dust from Sahara? How long do dust incursions last? This data should be shown in the Results section. It seems that they only last for a few days, so comparing inhaler sales over a month, rather than the days of the SDI, will make it more difficult to reveal any associations between SDI and inhaler sales.

We sincerely thank the reviewer for these constructive comments, which have helped us to clarify and strengthen the manuscript.

  1. Sources of PM₁₀ particles:
    We agree that sources other than Saharan dust may contribute to ambient PM₁₀ levels in the Canary Islands. Although long-range transport of mineral dust from the Sahara Desert (Calima events) constitutes the main source due to the archipelago’s geographical proximity to the African mainland, local contributions from road traffic, sea salt, and industrial emissions may also be present. Nevertheless, during Calima episodes, the Saharan dust component is predominant, as demonstrated by particle composition and back-trajectory analyses in previous studies.

  2. Duration of dust incursions and data presentation:
    As noted in the modified version of the manuscript, the Canary Islands experience approximately 24 Calima events per year, each lasting on average 1.8 days. These events are most frequent and intense from October to March, when low-altitude transport markedly increases PM concentrations at ground level. This information has now been explicitly included in the Introduction section (Line 101) to provide a clearer context for interpreting our findings.
  3. Regarding the temporal resolution, we acknowledge the reviewer’s concern. Monthly inhaler sales data were used due to the available temporal resolution in the pharmacy records, which allowed for the analysis of general patterns while minimizing random short-term variability. However, we agree that Saharan Dust Intrusions (SDIs) are short-lived phenomena, and this limitation is now discussed in the revised manuscript. Future studies incorporating daily or weekly medication dispensing data could provide a more detailed assessment of the short-term impact of SDIs on respiratory health outcomes.
    The following text has been added to the Discussion section (Line 362) to clarify this issue: “The relatively short duration of SDI (typically 1–2 days) compared with the monthly aggregation of pharmacy sales represents a methodological limitation that may attenuate the detection of short-term associations. Future research incorporating daily or weekly dispensing data could allow a more precise temporal alignment between dust events and changes in medication use.”

Could the associations in winter months be related to greater respiratory tract infections in Winter rather than SDI?

Answer: We sincerely appreciate the Reviewer’s remark. We agree that additional factors—such respiratory viral infections, and local airborne allergens—can influence both PM₁₀ concentrations and inhaler dispensations. We have now included a sentence in the Discussion section (limitations) acknowledging their potential influence (please, see below).

Line 404: Moreover, other factors may also influence PM₁₀ concentrations and inhaler dispensations, including local industrial and traffic emissions, variations in respiratory viral infections, and airborne allergen exposure. While in the Canary Islands the contribution of anthropogenic PM₁₀ sources is modest compared with Saharan dust intrusions [35,70], the potential impact of these environmental and biological cofactors should be considered when interpreting the observed associations.

ICS - formoterol inhalers can be used as relievers as well as maintenance treatment for asthma, whereas other types of ICS-LABA inhalers are only used for maintenance treatment of asthma. Do you have data separating these different types of ICS-LABA inhalers?

Answer: We thank the Reviewer for this important observation. We fully agree that inhalers containing inhaled corticosteroid–formoterol (ICS–formoterol) can be used both as maintenance and reliever therapy (MART/SMART regimen), whereas other ICS–LABA combinations are prescribed exclusively for maintenance treatment.

Unfortunately, in the present study, the aggregated pharmacy sales data did not allow differentiation between specific ICS–LABA formulations (e.g., formoterol vs. salmeterol or vilanterol). Consequently, it was not possible to identify which proportion of ICS–LABA inhalers was used as part of a MART regimen versus regular maintenance therapy. This limitation has now been explicitly acknowledged in the revised manuscript (line 280) as follows: “It should also be noted that pharmacy sales data did not allow differentiation between specific ICS–LABA formulations. As a result, we were unable to distinguish between ICS–formoterol inhalers, which may be used both for maintenance and reliever therapy, and other ICS–LABA combinations (e.g., ICS–salmeterol or ICS–vilanterol), which are prescribed exclusively for maintenance treatment [44-46]. This limitation should be considered when interpreting the ICS–LABA sales data.”

RESULTS I am concerned about the comment that higher SABA sales in GC reflect better  adherence with reliever medication. Could it not also mean that adherence with ICS-LABA is poorer, so exacerbations requiring SABA are more likely?

Answer: We thank the reviewer for this valuable comment and fully agree that higher short-acting β₂-agonist (SABA) sales could reflect different clinical or behavioural scenarios. Our interpretation that increased SABA sales in Gran Canaria might indicate better adherence to reliever medication was intended as one possible explanation within a complex context of medication use. We agree, however, that higher SABA consumption may also reflect poorer adherence to maintenance therapy with inhaled corticosteroid–long-acting β₂-agonist (ICS–LABA) combinations, leading to a higher frequency of exacerbations and therefore increased SABA use. This alternative interpretation has now been incorporated into the revised manuscript as follows (Line 378 (Discussion section)): “Additionally, the finding of higher SABA sales in Gran Canaria should be interpreted with caution. While this could partly reflect greater adherence to reliever therapy, it may also indicate suboptimal adherence to maintenance treatment with ICS–LABA combinations, leading to more frequent exacerbations and, consequently, greater reliance on SABA medication. Therefore, SABA dispensing data alone cannot distinguish between appropriate symptom-driven use and overuse related to poor controller adherence [62-64]. Another drawback relates to the inability to differentiate between specific ICS–LABA combinations in the aggregated pharmacy data. In particular, ICS–formoterol inhalers can be used both for maintenance and as-needed reliever therapy (MART/SMART regimen), whereas other ICS–LABA combinations are restricted to maintenance treatment.”

Round 2

Reviewer 2 Report

Comments and Suggestions for Authors

Thank you for your responses to the issues that I raised, and for the changes to the manuscript.